# Flipped Classroom as an Active Learning Methodology in Sustainable Development Curricula

**Marian Buil-Fabregá [1], Matilde Martínez Casanovas [1], Noemí Ruiz-Munzón [1]** 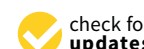 **and Walter Leal Filho [2,3],\***

1. Escola Superior de Ciències Socials i de l'Empresa Tecnocampus, Pompeu Fabra University, Mataró, 08302 Barcelona, Spain
2. European School of Sustainability Science and Research, Faculty of Life Sciences, Hamburg University of Applied Sciences, Ulmenliet 20, D-21033 Hamburg, Germany
3. Department of Natural Sciences, Manchester Metropolitan University, Manchester M1 5GD, UK
* Correspondence: walter.leal2@haw-hamburg.de

**Abstract:** Goal 4 of the Agenda 2030 sustainable development goals (SDGs) is aimed at working towards quality in education. Universities have an important role in teaching sustainability principles. Yet, which methods are effective for engaging students in understanding the importance of sustainable development and introducing them to new perspectives to make changes? The methodology of the flipped classroom is a possible alternative for the pedagogic renovation. This is known as an information-based environment in which teachers provide a variety of learning resources so that students can complete the knowledge transfer process before the class. Once inside classroom, teachers and students can complete the internalization of knowledge by answering questions, and through collaborative consultations and interactive exchanges, among others. A survey of 154 students taught by flipped classroom methodology was conducted in order to analyze whether this helps with learning about sustainable development. The results show the active and reflexive learning from flipped classroom methodology makes students more committed to sustainable development. This research would be useful to anyone interested in applying the flip the class teaching methodology as an integrated form of thinking and training in the curriculum of sustainable development for higher education students.

**Keywords:** sustainable development; flipped classroom; active learning methodology; sustainable development curricula

## 1. Introduction

### 1.1. Education for Sustainable Development

Teachers at higher education institutions have the continuous challenge of finding new methods to involve students in the classroom by increasing the effectiveness of the learning process, including sustainable development. The flipped classroom methodology reverses the normal learning process by moving the lectures out of the classroom and moving the concepts learnt in class through the use of learning activities [1]. Learning theorists argue that instructional strategies such as those used in an inverted classroom allow students to learn and retain information better than through traditional lectures [2–4].

Higher education institutions are increasingly recognized as a key drivers for the development of sustainable societies [5–8]. As Wals and Kieft [9] state, education for sustainable development (ESD) comes from environmental education (EE) and, for this reason, ESD has some elements in common

with EE. EE is focused on environmental concerns such as environmental protection, natural resource management, and conservation of nature. ESD is centered on humans bringing socioeconomic, political, and cultural dimensions to environmental concerns of EE. Gadotti [10] gives a suitable explanation in terms of similarities and differences between ESD and EE.

Recent research on ESD is paying attention on how to educate for sustainable development, giving a holistic approach to the sustainable development (SD) concept integrating the three dimensions of the term: environment, society, and economy [11]. The pedagogical dimension of ESD has taken an important role in ESD research, shifting from training and instructing on SD to learning, capacity-building, and participation for SD [12,13]. However, there are few studies trying to connect pedagogical approaches used in higher education and how they develop sustainability competences [14].

To date, most of the tools and methodologies used to measure and evaluate ESD come from indicators developed in the fields of education for citizenship, education for conservation and, in particular, EE [15,16]. McKeown and Hopkins [17] argued that ESD is not likely to replace EE but, rather, become one of its important objectives. EE evaluations sometimes include "sustainability" issues as evidenced by the volume on EE evaluations [18] and special journal numbers, e.g., *The Journal of Environmental Education* (1982), 13 (4), *New Directions for Evaluation* (2005), 108, *Journal of Evaluation and Program Planning* (2010), 33 (2), as well as online resources such as the Assessment Instrument for Sustainability in Higher Education (AISHE, 2014), and since it is argued that ESD assessments can benefit from the "lessons learned" from EE evaluations [19], comparing the objectives of EE and ESD seems justified.

On the other hand, one of the objectives of ESD is to devise the most efficient and strategic method to achieve the set objectives [20,21]. The most important challenge is didactic: What teaching methods are better considering the objectives of the ESD? Which of these methods make students involved and motivated? The authors propose to implement the flipped classroom approach since it facilitates critical thinking and improves participation both inside and outside the classroom. The results of implementing a successful flipped-class approach should consider students' effective learning that facilitates critical thinking [22] and, most importantly, improves student participation [23], both inside and outside the class.

## 1.2. The Flipped Classroom Approach

One of the simplest definitions of the flipped classroom is the one that comments what is usually done in class, at school, is now done at home and then, in class, the teacher completes it by helping students do what they would normally do at home [24–27].

According to the Flipped Learning Network, in 2014 (FLN 2014) the flipped classroom approach has four pillars. In order for teachers to achieve this approach, they have to take these four elements into consideration:

i.     Flipped learning requires flexible environments.
ii.    L: Flipped learning requires a shift in learning culture.
iii.   I: Flipped learning requires intentional content.
iv.    P: Flipped learning requires professional educators.

According to Bergmann, Overmyer, and Wilie [25], in the flipped classroom, the interaction time between the teacher and the student is increased, the students assume their own learning responsibilities, the role of the teacher evolves to a guide, there is a mixture of constructivist learning with the teaching method, and each student assumes their individual roles in education.

Two different roles are involved in the flipped classroom: the role of the teacher and the role of the student. The role of the teacher is the most important in this approach since, in the first place, instead of transferring knowledge directly, they must be a guide to facilitate learning [28]; second, they must create learning conditions based on questions [26]; they must also correct misunderstandings [26] and devise how to increase student participation [29]; additionally, it is necessary to have the appropriate

technological equipment to be able to learn [30] and to be able to share conference videos as out-of-class activities [31]; and feedback should be provided through the use of pedagogical strategies [32], among others. Meanwhile, the role of the student goes from being a passive receiver of knowledge to an active promoter of knowledge, assuming their own learning responsibilities [29], watching videos and using the learning materials before the class [33], participating in discussions within the class [34], and interacting with their teacher and friends, taking and giving feedback [35], as well as participating in teamwork [24].

As Quendler and Lamb [36] argued, ESD in combination with lifelong learning—which is part of the flipped classroom approach—combines the dimension of sustainability and the dimension of lifelong learning. In this sense, ESD is about learning to know, to do, to be and to live together. This means you need to put your knowledge into practice and one convenient methodology to apply is the flipped classroom encouraging students to be active learners, promoting participation and interaction towards a more sustainable world.

Figure 1 describes some of the characteristics of the flipped classroom model.

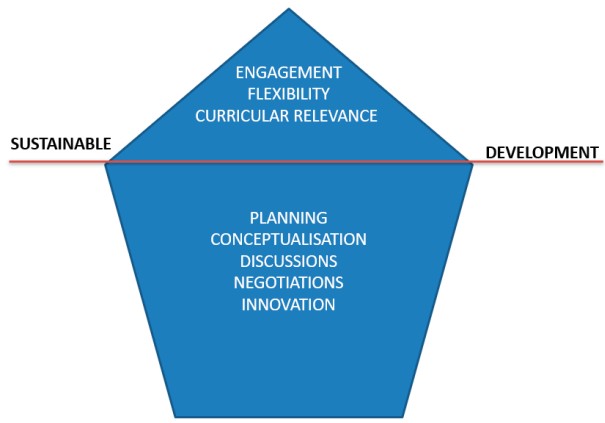

**Figure 1.** Some of the characteristics of the flipped classroom model.

Recent studies on the relevance of the application of active methodologies state ESD has shown considerable interest in the development of critical thinking (CT), which requires students to develop the necessary competencies. Consequently, a reorientation of higher education has become necessary. As such, active learning strategies have been introduced in the classroom to increase students' problem solving and critical thinking skills. There are numerous teaching methods to promote thinking and active learning in the classroom, including case studies, discussion methods, flipped classroom, questioning techniques, workshop, and debates, among others. Therefore, the consideration of active methodologies based on emerging pedagogies allows for improved student achievement and the development of competences, providing critical, significant, ubiquitous, transformative, and especially motivating experiences [37]. Moreover, it is confirmed that the flipped classroom acts as a transformative educational methodology allowing students to acquire competences which are part of the competences required for ESD. Some of these competences are also introduced in ESD to foster sustainable development goals (SDGs) as UNESCO stated in its learning objectives in education for SDGs.

## 2. Methodology

The objective of this research is to study whether the flipped classroom methodology could help in fostering learning about sustainable development issues in higher education. Perceptions of the learning processes of students who have been taught through a flipped classroom methodology during the period 2015–2019, and their commitment to SD, are analyzed. The chosen university is Tecnocampus, affiliated to Pompeu Fabra University in Barcelona (Spain). Tecnocampus has been studied as a start-up ecosystem development [38] and an innovation and entrepreneurship base

learning scientific park and university [39,40]. Tecnocampus is one of the 25 universities with good practice and case studies throughout Europe, chosen in 2016 by the "University–Business Cooperation in Europe", an initiative of the European Commission. Dr. Richard Woolley developed the case of Tecnocampus using an interview with Tecnocampus stakeholders.

Tecnocampus is an ecosystem where students, entrepreneurs, businesses, researchers, academics, and local government interact to share knowledge, contribute to sustainable regional economic development, and build successful futures. The co-located university faculties, start-up incubator, business park, and technological centers are connected through their common focus on entrepreneurship that is integral to all Tecnocampus education in business, health, and technology.

## 2.1. Procedure

During the years 2015–2019, the students received some of their classes through different active learning methodologies, among them, the flipped classroom. With this methodology, the students carried out individual and then group work, developing their materials through videos, presentations, readings, and reflections about the class. During the class, the teacher guided them through their individual and collective learning experiences.

The process of design and development of the flipped classroom was distributed as follows:

(1) Creation and preparation of contents: texts, videos, reference papers, search of contents, and resources of the network as well as texts and images of own development were used. Individual and collaborative activities were developed through Moodle, Office, and Drive.

(2) Development of the learning environment through the Moodle platform available 24 h a day, 7 days a week. The students were divided into groups so that once they advanced with the contents, they will carry out the assimilation work in a group way. In the first place, the students, as they progressed with the reading of contents, understood the lesson, viewing videos, images and answering the questions previously prepared by the teacher. Once the lesson was over, they were distributed in the previously assigned groups where they had to complement the questions in a group way and enrich the contents with new articles, videos, and images. Finally, forums were organized to discuss ideas and clarify doubts before arriving in class.

(3) During the face-to-face class, students first made presentations of their group work, analyzed their answers, and discussed them. By dividing the students into small groups, discussion and debate were encouraged and knowledge was deepened, especially through the examples they themselves brought and the problems they encountered. Then, the teams were reorganized for final reflections, encouraging collaborative learning and deepening the contents in the classroom. Finally, the students themselves made the final conclusions of the most relevant topics of the contents.

(4) To finish, the students developed a final work of the lesson in their own platform in which they unified the previous contents, the phase-to-face, and the final reflections in order to be ready for the final exam.

A pilot test of the survey was done in a focus group with 5 students from the target group. It was then observed how they completed the survey. Improvements were made to questions which were not clear enough.

## 2.2. Sample and Data Collection

The authors prepared a questionnaire that included adaptation of the flipped classroom quality assessment perception [41] and on the adaptation of the methodology section of the education module of AISHE 2.0 proposed by Roorda [42]. All the questionnaires were completed during the last fortnight of the month of March.

A total of 154 students who have participated in a flipped classroom methodology in the subject of creativity and innovation in two different grades of Business and Innovation Management (BaIM) and double Bachelor's degree in tourism and Leisure Management/BaIM (Db Tour) from the second, third,

fourth, and fifth courses during the last 3 years have responded to the questionnaire of the School of Business and Social Sciences. Of the total number of students surveyed, 45.45% were men and 54.55% women. There was a higher participation of men in BaIM with 60.67% compared to 39.33%, while in the Db Tour grade, the participation of women was higher, with 75.38% compared to a 24.62% of men. Finally Table 1 shows the differences in the distribution of students according to grade and year studied of the subject and work or not.

**Table 1.** Students' distribution according to grade and year studied of the subject and work or not.

| | Course 2018–2019 (0) | | Course 2017–2018 (1) | | Course 2016–2017 and Earlier (2) | |
|---|---|---|---|---|---|---|
| **Are You Working Now?** | **No** | **Yes** | **No** | **Yes** | **No** | **Yes** |
| BaIM | 34 | 8 | 9 | 11 | 6 | 21 |
| Db Tour | 19 | 4 | 7 | 13 | 5 | 17 |

*2.3. Methods for Evaluation of Flipped Classroom and Sustainability Development*

For this research, the authors have prepared a questionnaire based on two existing questionnaires. In the first place, to know the perception of the students who have participated in the flipped classroom learning methodology, they have adapted the questionnaire "perception of the quality of learning by the student", designed by Thomas Driscoll [41]. Second, to evaluate sustainability in higher education, the authors have made an adaptation of the AISHE 2.0 education module (Assessment Instrument for Sustainability in Higher Education). The details are explained below.

2.3.1. Flipped Classroom Quality Assessment Perception

This questionnaire aims to understand the perception of students who have participated in the learning process under a flipped classroom model. It addresses key issues in the teaching–learning process that are related to social, cognitive, and motivational variables. The questionnaire is structured on a Likert scale: "Strongly agree" (5)–"Agree" (4)–"Neither agreement nor disagreement" (3)–"Disagree" (2)–"Strongly disagree" (1). The survey questions are:

Q1: When I was working on my team paper-reading assignment, my communication with the professor was more frequent and positive.

Q2: When I was working on my team paper-reading assignment, my communication with my teammates was more frequent and positive.

Q3: Having been previously supplied with the necessary materials and contents contributed to my learning process.

Q4: Searching additional information to carry out my assignment allowed me to choose the kind of material that better suited my way of learning.

Q5: I had more chances to work at my own pace.

Q6: I had more chances to take part in the resolution of problems and develop my critical thoughts.

Q7: I had more chances to make decisions when I worked together with other teammates in the class.

Q8: I had more chances to show my professor and my teammates what I had learnt.

Q9: The learning process in the subject of Creativity and Innovation is more active and experimental than in other subjects based on traditional methodologies.

Q10: I think the professor is more likely to take into account my strengths, weaknesses, and interests.

2.3.2. Sustainability Assessment

For this research, the authors based their work on AISHE 2.0 (Assessment Instrument for Sustainability in Higher Education). This tool is based on a general method by the European Foundation for Quality Management (EFQM) and has been recognized as a standard for (self-)evaluation and

accreditation by the Dutch-Flemish Accreditation Organization. This new version is based on an EFQM quality approach, where each of the criteria is evaluated and qualified qualitatively. It also has a quantitative dimension where each criterion is evaluated on a scale between 0 (without SD integration) and 5 (complete, systemic, and society-oriented SD integration).

AISHE 2.0 was developed by an international development group (Niko Roorda, Christian Rammel, Sylvia Waara, and Urbano Fra Paleo) with a modular structure, which makes it possible to select those modules in which a university is interned. This modular structure is based on the 4 roles of universities in society, and for each role, there is a module (identity, operations, education, research, and society module).

For this study, the authors have focused on the methodology of the education module of AISHE 2.0. AISHE is an instrument designed specifically for evaluating sustainability in educational institutions [42,43]. These aspects were evaluated by students who have taken the subject of Creativity and Innovation with a flipped classroom methodology over the past 3 years. In this way, we can compare the opinions of the students over the years (immediately or over time). The authors have adapted the education module, including questions related to educational methodologies. The questionnaire is structured on a Likert scale: "Strongly agree" (5)–"Agree" (4)–"Neither agreement nor disagreement" (3)–"Disagree" (2)–"Strongly disagree" (1).

## 3. Results and Discussion

This section shows the main results of the analysis of the perception questionnaire given to students enrolled in flipped classroom methodology and the results of the same students regarding the AISHE education module for evaluating sustainability in educational institutions. Statistical analysis of the results was conducted using Excel and Minitab Statistical Software version 15 [44].

### 3.1. Results of Flipped Classroom Quality Assessment Perception

The analysis carried out during the attendance to the course Creativity and Innovation under the flipped classroom model shows an improvement in the perceptions of students. As seen in Figure 2 (BaIM), the perception of students who have finished the course (0) in the current course shows a neutral perception with a slight tendency to agree. The perception of the students who studied the subject a year before has increased significantly.

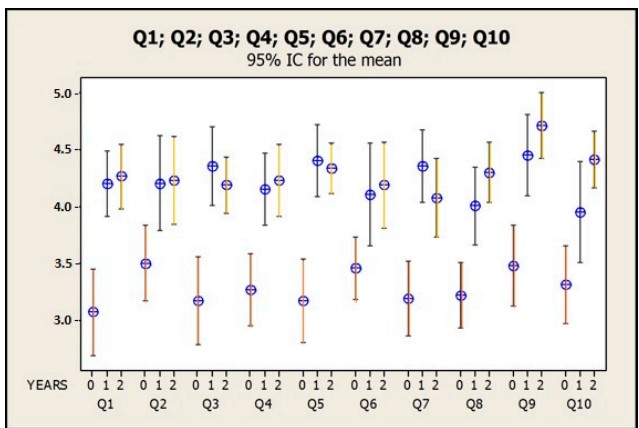

**Figure 2.** The evolution of perception according to long ago finished the subject (BaIM).

Table 2 of the ANOVA tests for each question corroborates the existence of significant differences between the averages of the different promotions.

**Table 2.** ANOVA test.

|  |  | Sum of Square | df | Mean Square | F | Sig. |
|---|---|---|---|---|---|---|
| **Q1** | YEAR | 30.020 | 2 | 15.010 | 15.90 | 0.000 |
|  | Within Groups | 81.171 | 86 | 0.944 |  |  |
|  | Total | 111.910 | 88 |  |  |  |
| **Q2** | YEAR | 11.270 | 2 | 5.64 | 5.61 | 0.050 |
|  | Within Groups | 86.370 | 86 | 1 |  |  |
|  | Total | 97.640 | 88 |  |  |  |
| **Q3** | YEAR | 26.599 | 2 | 13.299 | 13.87 | 0.000 |
|  | Within Groups | 82.457 | 86 | 0.959 |  |  |
|  | Total | 109.056 | 88 |  |  |  |
| **Q4** | YEAR | 19.226 | 2 | 9.613 | 12.28 | 0.000 |
|  | Within Groups | 67.336 | 86 | 0.783 |  |  |
|  | Total | 86.562 | 88 |  |  |  |
| **Q5** | YEAR | 31.726 | 2 | 15.863 | 18.78 | 0.000 |
|  | Within Groups | 72.633 | 86 |  |  |  |
|  | Total | 104.36 | 88 |  |  |  |
| **Q6** | YEAR | 10.845 | 2 | 5.422 | 6.28 | 0.003 |
|  | Within Groups | 74.279 | 86 | 1 |  |  |
|  | Total | 85.124 | 88 |  |  |  |
| **Q7** | YEAR | 23.099 | 2 | 11.550 | 13.63 | 0.000 |
|  | Within Groups | 72.878 | 86 | 0.847 |  |  |
|  | Total | 95.978 | 88 |  |  |  |
| **Q8** | YEAR | 21.276 | 2 | 10.638 | 16.14 | 0.000 |
|  | Within Groups | 56.701 | 86 | 0.659 |  |  |
|  | Total | 77.978 | 88 |  |  |  |
| **Q9** | YEAR | 28.540 | 2 | 14.270 | 15.93 | 0.000 |
|  | Within Groups | 77.056 | 86 | 0.896 |  |  |
|  | Total | 105.596 | 88 |  |  |  |
| **Q10** | YEAR | 20.499 | 2 | 10.250 | 11.53 | 0.000 |
|  | Within Groups | 76.445 | 86 | 1 |  |  |
|  | Total | 96.994 | 88 |  |  |  |

In Db Tour (Figure 3), a similar effect is observed. The perception of students who have finished the course in the current course (0) or the previous year (1) shows a neutral perception with a tendency to agree, while the perception of the students who studied the subject two years beforehand (2) increases significantly.

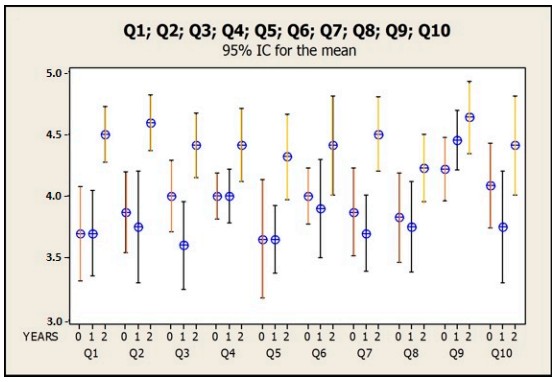

**Figure 3.** The evolution of perception according to long ago finished the subject (Db Tour).

Table 3 of the ANOVA tests for each question corroborates the existence of significant differences between the averages of the different promotions.

**Table 3.** ANOVA test.

|  |  | Sum of Square | df | Mean Square | F | Sig. |
|---|---|---|---|---|---|---|
| Q1 | YEAR | 9.369 | 2 | 4.684 | 9.82 | 0.000 |
|  | Within Groups | 32.570 | 62 | 0.525 |  |  |
|  | Total | 41.938 | 64 |  |  |  |
| Q2 | YEAR | 8.939 | 2 | 4.469 | 7.77 | 0.001 |
|  | Within Groups | 35.677 | 62 | 0.575 |  |  |
|  | Total | 44.615 | 64 |  |  |  |
| Q3 | YEAR | 6.866 | 2 | 3.433 | 7.57 | 0.001 |
|  | Within Groups | 28.118 | 62 | 0.454 |  |  |
|  | Total | 34.985 | 64 |  |  |  |
| Q4 | YEAR | 2.436 | 2 | 1.218 | 4.36 | 0.017 |
|  | Within Groups | 17.318 | 62 | 0.279 |  |  |
|  | Total | 19.754 | 64 |  |  |  |
| Q5 | YEAR | 6.475 | 2 | 3.238 | 4.31 | 0.018 |
|  | Within Groups | 46.54 | 62 | 1 |  |  |
|  | Total | 53.015 | 64 |  |  |  |
| Q6 | YEAR | 3.128 | 2 | 1.564 | 2.61 | 0.081 |
|  | Within Groups | 37.118 | 62 | 1 |  |  |
|  | Total | 40.246 | 64 |  |  |  |
| Q7 | YEAR | 7.63 | 2 | 3.815 | 7.32 | 0.001 |
|  | Within Groups | 32.309 | 62 | 0.521 |  |  |
|  | Total | 39.938 | 64 |  |  |  |
| Q8 | YEAR | 2.836 | 2 | 1.418 | 2.52 | 0.089 |
|  | Within Groups | 34.918 | 62 | 0.563 |  |  |
|  | Total | 37.754 | 64 |  |  |  |
| Q9 | YEAR | 1.985 | 2 | 0.992 | 2.80 | 0.068 |
|  | Within Groups | 21.954 | 62 | 0.354 |  |  |
|  | Total | 23.938 | 64 |  |  |  |
| Q10 | YEAR | 4.552 | 2 | 2.276 | 2.89 | 0.063 |
|  | Within Groups | 48.894 | 62 | 1 |  |  |
|  | Total | 53.446 | 64 |  |  |  |

In the case of BaIM, the results also show significant differences in all of the questions in the cases in which the students combined their studies with work practices or employment contracts. The perception of students under the flipped classroom model is valued higher compared to those who work on those who do not work. As we can see in Table 4 (calculated use software Minitab 15, assuming 95% confidence), the student's perception and communication with the teacher when preparing work under the flipped classroom method shows a greater degree of agreement in the cases in which the students are working or in an internship ($p < 0.05$).

**Table 4.** Two-Sample T-Test and CI: Q1_BaIM, "Work or not".

| Two-Sample T for Q1 vs. Work/Internship. | | | | |
|---|---|---|---|---|
| **Work/Internship** | **N** | **Mean** | **StDev** | **SE Mean** |
| 0 | 49 | 3.290 | 1.120 | 0.16 |
| 1 | 40 | 4.175 | 0.931 | 0.15 |

Difference = mu (0) − mu (1)
Estimate for difference: −0.889
T-Test of difference = 0 (vs. <): T-Value = −4.09 P-Value = 0.000 DF = 86

These results are aligned with Fulton [30], for whom the most important advantage of the flipped classroom approach is that the time within the class dedicated to the interaction between teacher and student is increased. As a result, the teacher can spend more time meeting student learning demands [45]. Moreover, in line with Milman [33], this study shows that the flipped classroom supports and improves teamwork within the classroom and gives higher education students more time to do novel research.

However, the factor of increasing satisfaction when being in the working world changes depending on the years when the course was completed, as shown by the analysis of multiple regression of Table 5. That is, a student of BaIM who works and has just finished the subject will score, on average, a Q1 of 4.023.

**Table 5.** Regression Analysis: Q2_BaIM vs. "years finished the subject", "Work or not", "years finished the subject", *"Work or not".

| The Regression Equation Is | | | | | |
|---|---|---|---|---|---|
| Q1 = 2.91 + 0.875 NYearsFinished + 1.12 Work/Internship −0.765 NYearsFinished*Work/Internship | | | | | |
| **Predictor** | **Coef** | **SE Coef** | **T** | **P** | |
| Constant | 2.9107 | 0.1575 | 18.49 | 0.000 | |
| NYearsFinished | 0.8750 | 0.1919 | 4.56 | 0.000 | |
| Work/Internship | 11.185 | 0.3309 | 3.38 | 0.001 | |
| NYearsFinished*Work/Internship | −0.7650 | 0.2692 | −2.84 | 0.006 | |
| S = 0.939964    R-Sq. = 32.5%    R-Sq.(adj) = 30.1% | | | | | |
| **Analysis of Variance** | | | | | |
| **Source** | **DF** | **SS** | **MS** | **F** | **P** |
| Regression | 3 | 36.091 | 12.030 | 13.62 | 0.000 |
| Residual Error | 85 | 75.100 | 0.884 | | |
| Total | 88 | 111.191 | | | |

The perception of the BaIM student and communication with his classmates when work is prepared under the flipped classroom method shows a greater degree of agreement in the cases when the students are employed or in internship (see Table 6, $p < 0.05$).

**Table 6.** Two-Sample T-Test and CI: Q2_BaIM, "Work or not".

| Two-Sample T for Q2 vs. Work/Internship. | | | | |
|---|---|---|---|---|
| **Work/Internship** | **N** | **Mean** | **StDev** | **SE Mean** |
| 0 | 49 | 3.670 | 1.110 | 0.16 |
| 1 | 40 | 4.125 | 0.939 | 0.15 |
| Difference = mu (0) − mu (1) | | | | |
| Estimate for difference: −0.452 | | | | |
| T-Test of difference = 0 (vs. <): T-Value = −2.08 P-Value = 0.020 DF = 86 | | | | |

These results provide evidence of the need for more interactions with teachers and friends and the need to take and provide feedback [35] as well as participating in teamwork [24]. Both ideas are part of collaborative learning for students. Collaborative education is the basis of education for sustainable development, as Roorda [42] (2008) states in AISHE.

The student's perception of having the materials and contents previously available when preparing work under the flipped classroom method shows a greater degree of agreement in the cases in which the students are employed or in an internship (see Table 7).

**Table 7.** Two-Sample T-Test and CI: Q3_BaIM, "Work or not".

| Two-Sample T for Q3 vs. Work/Internship | | | | |
|---|---|---|---|---|
| **Work/Internship** | **N** | **Mean** | **StDev** | **SE Mean** |
| 0 | 49 | 3.51 | 1.210 | 0.17 |
| 1 | 40 | 4.025 | 0.920 | 0.15 |
| Difference = mu (0) − mu (1) | | | | |
| Estimate for difference: −0.139 | | | | |
| T-Test of difference = 0 (vs. <): T-Value = −2.28 P-Value = 0.013 DF = 86 | | | | |

In order to allow students to have the content in advance, it is important to state that teachers have to prepare the materials very well so that students can follow the course, and may use high-quality videos, both of which require time and resources [46].

These results reinforce the fact that being able to have the materials and to view or access them as often and whenever desired, helps with learning. It allows students to establish their own content organization and manage it at their own pace [47], respecting the rhythm of learning of the students according to their individual needs. On the contrary, some students initially went to class without preparation.

Questions Q4, Q5, Q6, Q7, Q8, and Q9 show similar results to the previous ones for BaIM students. In all cases, the perception of the students shows a greater degree of agreement in the students of the 2017–2018 course and the previous courses of BaIM. In taking into account the degree of agreement with the student's strengths, weaknesses and interests, this percentage has improved over the years. For example, 75% of the students of the 2017–2018 course and 92.59% of the students of the previous BaIM courses are shown to agree with Q10, as we can see in Table 8. In the case of Db Tour, 70% of the students of the 2017–2018 academic year and 90.91% of the students of the previous courses show a degree of agreement with Q10.

**Table 8.** Q10: I think the professor is more likely to take into account my strengths, weaknesses, and interests.

| | Strongly Disagree (1) | | Disagree (2) | | Neither Agreement nor Disagreement (3) | | Agree (4) | | Strongly Agree (5) | |
|---|---|---|---|---|---|---|---|---|---|---|
| **Are You Working Now?** | **No** | **Yes** | **No** | **Yes** | **No** | **Yes** | **No** | **Yes** | **No** | **Yes** |
| **BaIM** | | | | | | | | | | |
| Course 2018–2019 (0) | 9.52% | 2.38% | 4.76% | 0.00% | 26.19% | 4.76% | 33.33% | 11.90% | 7.14% | 0.00% |
| Course 2017–2018 (1) | 0.00% | 0.00% | 5.00% | 5.00% | 10.00% | 5.00% | 20.00% | 25.00% | 10.00% | 20.00% |
| Course 2016–2017 and earlier (2) | 0.00% | 0.00% | 0.00% | 0.00% | 7.41% | 0.00% | 3.70% | 40.74% | 11.11% | 37.04% |
| **Db Tour** | | | | | | | | | | |
| Course 2018–2019 (0) | 0.00% | 0.00% | 4.35% | 0.00% | 13.04% | 0.00% | 43.48% | 8.70% | 21.74% | 8.70% |
| Course 2017–2018 (1) | 0.00% | 0.00% | 10.00% | 5.00% | 5.00% | 10.00% | 15.00% | 35.00% | 5.00% | 15.00% |
| Course 2016–2017 and earlier (2) | 0.00% | 0.00% | 9.09% | 0.00% | 0.00% | 0.00% | 0.00% | 31.82% | 13.64% | 45.45% |

These results reinforce the idea of scenarios of "real-time feedback" [48]. It concludes that, as a teaching tool, immediate feedback is very beneficial for student learning. The immediate feedback in

the classroom, processed by the student and commented upon by the teacher, brings new knowledge to the student.

Finally, the data of the students of Db Tour did not take into account the effect of the student's contact with the working world. It would therefore be necessary to consider other variables and analyze the differences in the curriculum of the two grades more deeply.

### 3.2. Results in Relation to AISHE's Education Module for Evaluating Sustainability in Educational Institutions

The analysis carried out according to SD with the AISHE instrument shows the results from the median of each criterion in relation to the points attributed to the total sum of the evaluated courses. Figure 4 shows that the criterion educational methodology (2.1) showed the best results, reaching Stage 2, which signifies that the teaching and learning methodology gives the student the opportunity to encounter real situations, which gives rise both to reflection and development of his/her future professional development in a sustainable vision.

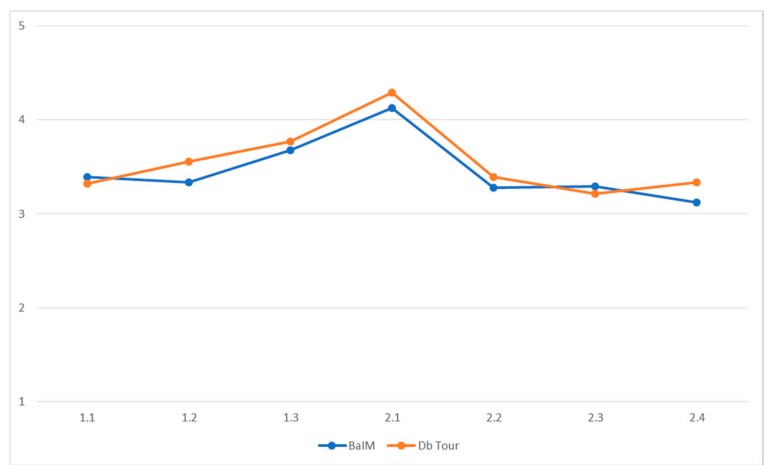

**Figure 4.** Results of the AISHE goals and methodology module.

The questions when asking about goals (1.3) in the education module of AISHE are as follows:

*"Organization demonstrably contributes to sustainable development on the level of adaptations and improvements.*

*Sustainable development in the profile is explicitly based on the vision of the organization about sustainable development.*

*Systematic evaluations and adjustments of the profile take place.*

*The profile explicitly demands multidisciplinary capacities."* [49]

The results show values higher than 3 in more than 50% of the answers (Table 9), with 67.94% of BaIM students agreeing with the goals of AISHE 1.3 education module and 64.70% in the case of Db Tour students. These results indicate that those students following a flipped classroom methodology seem to be more alert to SD development issues or, in other words, they are more committed to sustainability. This perception will be expanded in their future work helping to make improvements in sustainability as they agree that sustainable development should be part of the vision of companies.

**Table 9.** AISHE education module, goals, Stage 3, system-oriented.

| AISHE 1.3. | Strongly Disagree (1–1.99) | Disagree (2–2.99) | Neither Agree nor Disagree (3–3.99) | Agree (4–4.99) | Strongly Agree (5) |
|---|---|---|---|---|---|
| BaIM | 1.07% | 6.56% | 24.43% | 58.78% | 9.16% |
| Db Tour | 0.00% | 2.65% | 32.66% | 50.41% | 14.29% |

The results when asking about methodology (2.1) in the education module of AISHE are taken from the question:

*"In some parts of the curriculum, methodologies are used to stimulate some aspects of action learning and reflexivity."* [49]

The results indicate a degree of agreement slightly higher than 90%. The 91.30% of BaIM students agree or strongly agree on this aspect and 92.50% in the case of Db Tour (Table 10).

**Table 10.** AISHE education module, methodology, Stage 1, activity-oriented.

| AISHE 2.1 | Strongly Disagree (1) | Disagree (2) | Neither Agree nor Disagree (3) | Agree (4) | Strongly Agree (5) |
|---|---|---|---|---|---|
| BaIM | 0.27% | 2.72% | 5.72% | 49.05% | 42.23% |
| Db Tour | 0.00% | 0.00% | 7.53% | 45.88% | 46.59% |

These results reinforced the idea that active learning and reflexivity is part of learning about SD, as Quendler and Lamb [36] pointed out.

The results when asking about methodology (2.2) in the education module [49]:

*"In many parts of the curriculum, methodologies are used to stimulate many aspects of action learning and reflexive learning.*

*The methodologies have been selected in such a way that innovatively is stimulated."*

It is noted that 53.86% of the students of BaIM agree or strongly agree on this aspect, as do 56.69% of the students of the Db Tour (Table 11). These results also show the importance of active learning and reflexivity learning in the curricula of SD subjects which is part of flipped classroom methodology. Both types of learning could facilitate students to be more innovative when proposing solutions for sustainable development (Littledyke, Manolas, Littledyke 2013) in line with the SDGs.

**Table 11.** AISHE education module, methodology, Stage 2, process-oriented.

| AISHE 2.2 | Strongly Disagree (1–1.99) | Disagree (2–2.99) | Neither Agree nor Disagree (3–3.99) | Agree (4–4.99) | Strongly Agree (5) |
|---|---|---|---|---|---|
| BaIM | 1.54% | 18.18% | 26.42% | 48.71% | 5.15% |
| Db Tour | 0.45% | 16.55% | 26.31% | 52.15% | 4.54% |

Qualitative results were collected during the survey, providing a space to register any comment regarding the methodology used. These comments were collected as qualitative data and, in the results, some interesting comments to be considered for future research were "teamwork was present in most of the proposed activities", "practical learning method, I like it", "I'm conscious about what I have learnt in the topic", " practical and dynamic methodology", "expositions and debates where sharing information clarify concepts", among others.

## 4. Conclusions

The main contribution of this research is to demonstrate that the flipped classroom methodology is a good approach to use in SD curricula in order to make students conscious about SD requirements. The study shows that the active and reflexive learning taught to students in flipped classroom methodology are the items which seem to be connected with the SD commitment gained by students. Some differences on question 2.2 of the AISHE survey results were found, as to whether the methodologies used increased creativity and innovation skills which are essential for ESD.

Based on this research, there are several recommendations for higher education institutions and practitioners when using the flipped classroom model to help students learn about sustainable development. First, it is considered important to use innovative approaches in education. Although recent years have seen increased research on flipped classrooms, it is still not considered a well-known approach. To expand the use of flipped classrooms in higher education approaches, teacher skills must be developed both in the design and/or transforming of their materials and in the use of technological equipment to disseminate this approach. Second, it is recommended to continue polling students over the years on flipped classroom methodology and SD in order check if their perceptions have changed. This study provides evidence that these perceptions increase over the years as the active learning methodology was implemented from 2015 until the actual course 2019. Third, as employers are beginning to be aware of the need for training in SD, higher education institutions should include SD content in their curricula using methodologies which can enable learning in a more appropriate way. Finally, the research recommends that universities use the flipped classroom methodology to reach reflexive learning, which requires commitment to SD and the improvement of transversal competences of higher education students. Moreover, the flipped class methodology in sustainable development education helps to increase the commitment to sustainable development in the labor market, as the results show it is higher in students who are working.

There are some limitations to be taken into account. First, this paper is based on flipped classroom methodology as a methodology for sustainability, but there are more active learning methodologies which have the same purpose. The results when using other active learning methodologies could vary, and further studies using other methodologies should be conducted. Second, the sample responds to a concrete higher education institution, and it is possible that the same survey done in another university would not show the same results. Third, the study is based solely on the measurement of perceptions of students, and the measurement of perceptions of professors should be also conducted to have both perceptions of flipped class methodology.

The present study provides a base for further research. In line with other learning methodologies, it could be interesting to analyze which results help with the learning of SD when using other active methodologies. Another proposal is to study the relation between methodologies and competences for ESD. It could be also interesting to find other competences which could be important to foster sustainable development through SDGs. Finally, exploring differences related to gender and/or field of study regarding this topic would represent an innovative line of research.

**Author Contributions:** M.B.-F. and M.M.C. conceived the methodology. N.R.-M. made the data processing, M.B.-F., M.M.C. and N.R.-M. the analysis of them and M.B.-F., M.M.C., N.R.-M. and W.L.F. the writing of the paper.

**Funding:** This research received no external funding.

**Conflicts of Interest:** The authors declare no conflict of interest.

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
