# Peer review of "Flipped Classroom as an Active Learning Methodology in Sustainable Development Curricula"

_sustainability, doi:10.3390/su11174577_

Round 1

Reviewer 1 Report

The article presents, as a whole, an adequate structure, design and development, although it would not be necessary at this time to explain questions such as the meaning of an inverted classroom and to focus more on the relevance of the application of active methodologies and the value for the competence development of the student body as a whole. This would need to be corrected. It is also advisable to cite a recent publication in this journal that is closely linked to the study carried out:

Hinojo Lucena, F. J., Mingorance Estrada, A. C., Trujillo Torres, J. M., Aznar Díaz, I. and Cáceres Reche, M. P. (2018). Incidence of the flipped classroom in the physical education students' academic performance in university contexts. In Sustainability. Special Issue "Physical Activity as a Means of Culture, Leisure and Free Time". 

The value of statistical analysis resides in its concretion and correction in the applied set. 

Therefore, with the corrections in the theoretical framework linked by the authors themselves as limitations of the study itself, I recommend the publication of the manuscript. 

Best regards

Translated with www.DeepL.com/Translator

Reviewer 2 Report

The topic of the paper is very interested and strongly related to new teaching – learning approaches and methods in sustainable education in higher education.

There are several aspects in the paper that need to be addressed and revised.

The study is highly contextual (the sample is from just one university, participants being recruited from only two study programs and it seem that the researchers are from the same university). 

The issue of contextuality and specificity of the sample and of the results should be better addressed, mainly in the Methodology and Conclusion sections. 

In the Methodology section, it would be useful to present, in short, how flipped classroom methodology was implemented to the students.

In the Conclusion section, the results should be discussed in more details regarding to the specificity of the sample. 

The authors should state more clearly the limitations of the study based on their double role to the sample, on the solely measurement of perceptions of students. 

In the Conclusion section, the relevance of the flipped classroom methodology in sustainable education should be emphasized,  especially since the result show differences in favour of students who work.

In the Result section, the comparative analyses between groups is quite relevant, the regression analysis, also. However, there are several results that could be presented better. For example, the authors presented descriptive results for several questions from the questionnaire, maybe it would be more valuable to integrate answers from multiple questions and to use inferential methods of data analysis (like correlations or mean differences).

Also, information presented in figure 1 and 2 and in table 6 does not contribute significantly to the understanding of the data. It is expected that perceptions will change after engagement in any type of new approach, but it is not clear if the difference is significant or not.

The authors should revise the language. There are few typos and few unclear sentences. 

Reviewer 3 Report

* Line 48      ESD is first mention and wasn't presented to readers in full form...confusing

* Line 101-108   solid explication and literature on what FLIP entails

* Line 115   good point as both teachers and students will undergo a role change to respectively, a facilitator and active learner in flipped classroom methodology...namely what line 116-129 discuss

* Line 161   which year?

* Was the adapted survey instrument pilot-tested?  What is the reliability of the adapted instrument?  This information is critically from a methodological perspective because it lets readers know how valid and reliable is the researcher-developed instrument.

* Were some qualitative or text-based data collected during the process?  If yes the results would be quite reinforced when data triangulation was conducted

Round 2

Reviewer 2 Report

Dear Authors, The improvement work you have given to your manuscript in this latest version is very appreciable. In my opinion, the changes introduced in the manuscript enhanced substantially its scientific soundness and overall quality. That said, I have only a minor concerns, regarding general spect of the paper. Good work!

Author Response

revised